# MULTI-MODAL PROMPT LEARNING EMPOWERS GRAPH NEURAL NETWORKS WITH SEMANTIC KNOWLEDGE

## ABSTRACT

While great success has been achieved in building generalizable language models, three fundamental issues hinder GNN-based graph foundation models: the scarcity of labeled data, different levels of downstream tasks, and the conceptual gaps between domains. In depth, though the labels of real graphs are associated with semantic information, most graph learning frameworks ignore it by turning semantic labels into numerical labels. In this work, to address these issues, we present a new paradigm that leverages the text modality to align downstream tasks and data with any pre-trained GNN given only a few semantically labeled samples. Our paradigm embeds the graphs directly in the same space as the LLM by learning both graph prompts and text prompts simultaneously. To accomplish this, we improve state-of-the-art graph prompt method based on our theoretical findings. Then, we propose the first multi-modal prompt learning approach for exploiting the knowledge in pre-trained models. Notably, in our paradigm, the pre-trained GNN and the LLM are kept frozen, so the number of learnable parameters is much smaller than fine-tuning any pre-trained model. Through extensive experiments on real-world datasets, we demonstrate the superior performance of our paradigm in few-shot, multi-task-level, and cross-domain settings. Moreover, we build the first zero-shot classification prototype that can generalize GNNs to unseen classes. The code is provided in the supplementary materials.

## 1 INTRODUCTION

Foundation Models [4] learn generalizable representations from large-scale data and can be adapted to a wide range of downstream tasks. Although foundation models have shown remarkable capability and been thriving in NLP [14, 7, 112, 80], computer vision [2, 15, 65, 66, 44, 84], and time-series analysis [83, 110, 49], graph-related foundation models still remain in a very nascent stage. This is due to the significant difference of non-euclidean graph data from other data types. First, compared with language or vision data, graph data is very scarce [50, 10, 59] for foundation models. Second, the task space of graph data could be on node-level [86], edge-level [71], and graph-level [67]. Third, in general, language tokens and visual objects retain the same conceptual meaning across different distributions, but the same graph structure may have distinct interpretations in different domains, depending on how graphs were constructed from real scenarios. Thus, even if we have a pre-trained model, adapting it to various downstream tasks is not trivial.

Recently, some works [98, 8, 3, 82] reformulate the graphs into natural language descriptions and the graph tasks into natural language prompts, then query LLMs to generate the answer. However, since the LLMs are not directly trained from structured graph data [52], it is uncertain how LLMs could correctly solve those tasks without hallucinating [1, 31, 95, 105]. Nevertheless, graph neural networks (GNNs) are well-studied architectures for learning graph data [90, 17, 106], with theoretically provable expressiveness [94, 69, 61], better interpretability [13, 34, 74] and experimentally outstanding performance [89, 40]. Therefore, GNNs are expected to leverage their inherent advances for structure learning and inference on graphs in the era of big data and foundation models.

However, though tremendous efforts have been devoted to pre-train GNNs through self-supervision [92, 29, 54], a key problem in building a GNN-backboned graph foundation model is that GNNs do not capture semantics, given that current GNNs are optimized according to numerical labels. In other words, GNNs do not really *understand* what a label represents in the real world, even though the graphs are constructed from real scenarios. To solve the issue of predetermined numerical categories, CLIP [65] leverages natural language supervision by jointly training an image encoder and a text encoder in the same embedding space to predict the correct image-text pairs at scale. The excellent generalization ability of pre-trained V-L models [65, 33, 47] comes from the alignment between

the vision and language representations. Notably, some works have explored prompt learning for better alignment and obtained improvement in vision prediction [114, 41]. The idea of alignment with text modality has also been applied in video [93, 6], 3D images [103, 25, 26], speech [70] and audio [23, 85] areas. As for graphs, so far such CLIP pipelines have only be applied in the molecular domain [58, 56, 72, 51], where the paired graph-text data are relatively sufficient for pre-training to align representations. But for other domains, such text-labeled graph data are rarely available, which means we have to rely more on self-supervised GNN pre-training to build graph foundation models. With this assumption, it is necessary to study how to make the pre-trained GNN aware of the semantics of downstream graph representations, which motivates the following question:

*How to adapt pre-trained GNNs to the semantic embedding space given limited downstream data?*

This paper aims to answer this question based on the following observations: (1) Semantic text embedding spaces do not necessarily result from joint pre-training. In fact, the embedding spaces of encoder LLMs are inherently semantic and high-quality, as LLMs are trained on massive text data and demonstrate strong reasoning performance. (2) When the downstream data are limited, prompt learning [48, 28, 102, 45] provides a better option than fine-tuning as much fewer parameters not only makes the optimization more efficient but also requires less resource than computing the gradient of a large model. Inspired by these two observations, we propose a prompting-based paradigm with an LLM that, while keeping the parameters of both GNN and LLM frozen, aligns the GNN representations in the LLM's semantic embedding space.

Notably, when attempting to adapt the representation from one modality to another, solely prompting a single modality could be sub-optimal, as it limits the adjustment to downstream tasks in the other modality [41]. To this end, we propose Multi-modal Prompt Learning for Graph Neural Networks (Morpher). Given a pre-trained GNN and few-shot semantically labeled graph data, we introduce a pre-trained LLM. Then, to leverage its high-quality semantic embedding space, Morpher connects and aligns the graph embeddings to it through prompting on both modalities with a cross-modal projector. Nonetheless, designing such a paradigm is more challenging than vision-language models. First, we lack jointly pre-trained encoders for the two modalities; instead, we only have two encoders whose embedding dimension is possibly different, pre-trained independently in each modality. Second, determining how to prompt the graph modality is non-trivial and remains a trending research topic. Third, the downstream data for GNN usually have much fewer labeled classes than V-L models, so in the few-shot setting, the available downstream data is extremely limited. Our contributions towards tackling these challenges are summarized as follows:

- Theoretically, we analyze that, in many cases, state-of-the-art graph prompt [76] is unable to learn good representations of the downstream data. We show that the optimization of the graph prompt is restricted by design. From the theoretical findings, we further improve state-of-the-art graph prompt according to the attention mechanism to prevent failure in optimization.

- To connect and adapt the pre-trained GNN with LLM, we propose Morpher, a graph-text multi-modal prompt learning paradigm. To the best of our knowledge, this is the first approach to align the representations of GNN and LLM without fine-tuning any of their parameters.

- Experimentally, we demonstrate the effectiveness of our improved graph prompt and Morpher on real-world datasets under few-shot, multi-task, and cross-domain settings.

## 2 BACKGROUND

We use calligraphic letters (e.g., $\mathcal{A}$) for sets, and specifically $\mathcal{G}$ for graphs. We use bold capital letters for matrices (e.g., $\mathbf{A}$). For matrix indices, we use $\mathbf{A}(i, j)$ to denote the entry in the $i^{th}$ row and the $j^{th}$ column. Additionally, $\mathbf{A}(i, :)$ returns the $i^{th}$ row in $\mathbf{A}$.

**Graph Neural Networks.** We use $\mathcal{G} = (\mathbf{A}, \mathbf{X})$ to denote a graph with node set $\mathcal{V}$ and edge set $\mathcal{E}$, where $\mathbf{A} \in \mathbb{R}^{|\mathcal{V}| \times |\mathcal{V}|}$ is the adjacency matrix and $\mathbf{X} \in \mathbb{R}^{|\mathcal{V}| \times d}$ is the node feature matrix. $\mathbf{A}(u, v) = 1$ if there is an edge connecting $u$ and $v$; otherwise $\mathbf{A}(u, v) = 0$. A Graph Neural Network $f_\phi^g(\cdot)$ with hidden dimension $d_g$ encodes $\mathcal{G}$ into the embedding space: $f_\phi^g(\mathcal{G}) \in \mathbb{R}^{|\mathcal{V}| \times d_g}$, which could preserve both feature and structure information of $\mathcal{G}$. The extracted embeddings $f_\phi^g(\mathcal{G})$ can be used for various downstream tasks such as classification. Nowadays, a popular paradigm to train GNNs is to first pre-train GNNs via self-supervised learning [29] and then fine-tune on the downstream tasks.

**Few-shot Prompt Learning.** Prompt learning adds learnable tokens to the downstream data and provides a powerful alternative to fine-tuning when the labeled downstream data is scarce. Prompt learning for encoders was first used in NLP. Let $f_\phi^t(\cdot)$ denote the LLM encoder with embedding dimension $d_t$. For a series of input tokens $\{x_k\}_{k=1}^K$, the LLM encoder embeds it as a matrix $\mathbf{X}_t = f_\phi^t(\{x_k\}_{k=1}^K) \in \mathbb{R}^{K \times d_t}$, then aggregates the representation to a vector $aggre(\mathbf{X}_t) \in \mathbb{R}^{1 \times d_t}$ for downstream tasks. Prompt learning initializes a tunable matrix $\mathbf{P}_\theta^t \in \mathbb{R}^{n_t \times d_t}$, where $n_t$ denotes the number of text prompt tokens. Then, this tunable matrix is concatenated with the input tokens' embeddings to form a single matrix $[\mathbf{P}_\theta^t; \mathbf{X}_t]_{dim=0} \in \mathbb{R}^{(K+n_t) \times d_t}$, and the aggregated vector for downstream tasks becomes $aggre([\mathbf{P}_\theta^t; \mathbf{X}_t]_{dim=0})$. In practice, we can train the model to minimize the loss function for downstream tasks, with only the prompt parameters $\mathbf{P}_\theta^t$ being updated.

Now we are ready to introduce the problem setup for this work. Given a pre-trained GNN $f_\phi^g(\cdot)$ with embedding dimension $d_g$ and a pre-train LLM encoder $f_\phi^t(\cdot)$ with embedding dimension $d_t$. Without loss of generality, we assume the downstream task is graph-level classification, as we will show that the other types of GNN tasks can be reformulated as graph classification. For $L$-shot graph classification, we are given limited text-labeled pairs $\{(\mathcal{G}_i, t_c)\}_{i=1}^L$ for each class $c$. Assuming $\mathcal{T}$ is the set of all text labels $t_c$, we are provided a set of test graphs $\{\mathcal{G}_j\}_{j=1}^{L_{test}}$. Using the pre-trained GNN and LLM, we want to correctly predict the text label $t_j \in \mathcal{T}$ for each test graph $\mathcal{G}_j$.

## 3  REVISITING AND IMPROVING PROMPT AS GRAPHS

Unlike prompting text data (which can be easily achieved by appending learnable text tokens to the original text sequence) and prompting image data (which pads a learnable image area above the original image), prompting graph data presents a significant challenge due to the non-euclidean nature of graphs. The recent pioneering work [76] designs the graph prompt still as a graph, then inserts it into the original graph by computing the inner-connections within the prompt graph and the cross-connections between the prompt graph and the original graph. An advantage of prompting at the graph level is that *the downstream tasks of GNN can be reformulated into graph-level tasks*. For the node classification task, we can induce the $\gamma$-ego-graph of each node by extracting the subgraph within a pre-defined distance $\gamma$. Then, we treat the node label as the induced ego-graph label. Similarly, for the edge classification task, we can extract a subgraph for each edge by extending the node pair to their $\gamma$ distance neighborhood, and use the edge label as the induced graph label. By inducing subgraphs, we can reformulate node-level and edge-level downstream tasks to graph-level.

**Current Graph Prompt Design.** To prompt a graph $\mathcal{G}$, each prompt token is a new node. Let $n_g$ denote the number of prompt tokens and $\mathcal{P} = \{p_i\}_{i=1}^{n_g}$ denote the set of prompt tokens. The graph prompt is formulated by a tunable matrix $\mathbf{P}_\theta^g \in \mathbb{R}^{n_g \times d}$, where $d$ is the node feature dimension of graph $\mathcal{G}$. In other words, each row vector $\mathbf{P}_\theta^g(i,:)$ is the feature of the prompt token $p_i$. Then, the mechanism to prompt a graph $\mathcal{G} = (\mathbf{A}, \mathbf{X})$ with $n$ nodes and $d$ feature dimension is as follows [76].

- Compute inner-connections to construct the prompt graph $\mathcal{G}_p = (\mathbf{A}_p, \mathbf{X}_p)$. For the feature matrix, we directly set $\mathbf{X}_p = \mathbf{P}_\theta^g$. For two prompt tokens $p_i$ and $p_j$, the prompt graph will have an edge between them if and only if the dot product of their features is larger than a threshold. In other words, $\mathbf{A}_p(i,j) = 1 \iff \sigma(\mathbf{P}_\theta^g(i,:)\mathbf{P}_\theta^g(j,:)^\top) > \delta_{inner}$, where $\sigma(\cdot)$ is the sigmoid function.
- Compute cross-connections to insert the prompt graph $\mathcal{G}_p$ into the original input graph $\mathcal{G}$. Similarly, for $x_i \in \mathcal{G}$ and $p_j \in \mathcal{G}_p$, there is an edge between them if and only if $\sigma(\mathbf{X}(i,:)\mathbf{P}_\theta^g(j,:)^\top) > \delta_{cross}$.
- Construct the prompted graph (i.e., manipulated graph) $\mathcal{G}_m = (\mathbf{A}_m, \mathbf{X}_m)$. The overall adjacency matrix $\mathbf{A}_m \in \mathbb{R}^{(n+n_g) \times (n+n_g)}$ is constructed from the original adjacency matrix $\mathbf{A}$, the inner edges $\mathbf{A}_p$ and the cross edges. The overall node feature matrix is concatenated from the prompt token features and the original input node features: $\mathbf{X}_m = [\mathbf{P}_\theta^g; \mathbf{X}]_{dim=0} \in \mathbb{R}^{(n+n_g) \times d}$.

Here, we identify an issue associated with the current design. Since not all the GNN backbones can take edge weights [21], the cross-connections in a manipulated graph are discrete[1], thresholded by $\delta_{cross}$. However, the input node features of most real-world datasets are sparse, resulting from the construction process [97, 60, 18]. As shown in Table 6, $\|\mathbf{X}(i,:)\|_1$ is typically 1. As the initialization of each token feature $\mathbf{P}_\theta^g(i,:)$ is close to $\vec{\mathbf{0}}$, for any node $i$ and token $p_j$, the dot products

---

[1]In official implementation of [76], adjacency matrices are discrete: either 0 or 1 for each entry.

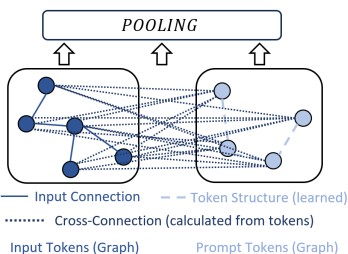 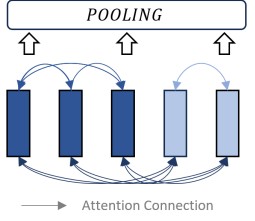 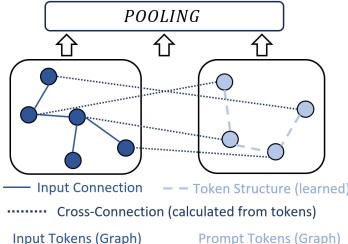

| Input Connection | — — Token Structure (learned) |
|---|---|
| ········ Cross-Connection (calculated from tokens) | |
| Input Tokens (Graph) | Prompt Tokens (Graph) |

| ——→ Attention Connection |
|---|
| Input Tokens (Text) | Prompt Tokens (Text) |

| Input Connection | — — Token Structure (learned) |
|---|---|
| ········ Cross-Connection (calculated from tokens) | |
| Input Tokens (Graph) | Prompt Tokens (Graph) |

Figure 1: Illustration of connections in current problematic graph prompt design (left), transformer architecture (middle), and our improved graph prompt design (right). **The cross-connections between input and prompt should be consistent with the input connections in scale.**

$\mathbf{X}(i,:)\mathbf{P}_\theta^g(j,:)^\top$ is close to $0$ , and the sigmoid value is very close to $0.5$. As a result, if we want the graph prompt to work reasonably, we have to set $\delta_{cross} < 0.5$. However, in this case, the cross-connections will be dense, i.e., almost every node in the original graph is connected with every node token in the prompt graph. For two different graphs $\mathcal{G}_1$ and $\mathcal{G}_2$ in the same task, the prompt graph $\mathcal{G}_p$ is identical. Since the GNNs work by aggregating the node features, their embeddings $f_\phi^g(\mathcal{G}_1)$ and $f_\phi^g(\mathcal{G}_2)$ are approximately the same because the features in the prompt graph overwhelm the features in the original graphs due to the dense cross-connections. Then, according to the following lemma, even if $\mathcal{G}_1$ and $\mathcal{G}_2$ have different labels, the task head classifier cannot be trained to distinguish them[2].

**Lemma 3.1.** *For any classifier $c(\cdot)$, if the identical feature $\mathbf{x}$ has label distribution $p(\cdot)$, then the optimal classification for cross-entropy loss is $Pr(c(\mathbf{x}) = y) = p(y)$. From this, if two graphs have similar embedding but different labels, GNN training may not converge. (Proof in Appendix A)*

**Improved Graph Prompt Design.** The issue of the current graph prompt is rooted in the imbalance of original connections in the input graph and cross-connections between input and prompt, as shown in Figure 1 (left). Since the text prompt works well in NLP, we look into the standard transformer architecture [81], where the token features are aggregated through the attention mechanism:

$$\widetilde{\mathbf{H}} = \text{Attn}_{\boldsymbol{\theta}_a}(\mathbf{H}) := \mathbf{H} + \frac{1}{N}\sum_{m=1}^{M}(\mathbf{V}_m\mathbf{H}) \times \sigma\big((\mathbf{Q}_m\mathbf{H})^\top(\mathbf{K}_m\mathbf{H})\big) \in \mathbb{R}^{D\times N} \tag{1}$$

where $\mathbf{H} \in \mathbb{R}^{D\times N}$ is the input sequence and $\boldsymbol{\theta}_a = \{(\mathbf{V}_m, \mathbf{Q}_m, \mathbf{K}_m)\}_{m\in[M]} \subset \mathbb{R}^{D\times D}$ denotes the parameters with $M$ heads. $N$ is number of tokens and $D$ is embedding dimension. We also visualize such attention mechanism in Figure 1 (middle). After we prepend a sequence of text prompt tokens $\{p_i^t\}$, the features of the text prompt tokens will be densely aggregated to the features of the original text tokens. In other words, the "cross-connection" between the text prompt sequence and the input sequence is dense. However, such a dense connection does not cause the prompt feature to overwhelm the input, because the features in the input sequence are also aggregated in a dense manner. Inspired by this, the number of cross-attention between input and prompt should approximate the number of input connections. Since the connection of a graph dataset is often sparse, we should also constrain the cross-connections between the prompt graph and the input graph to be sparse as well.

Nonetheless, "sparse" is a wide concept to implement: if the cross-connections are too dense, the prompt graph will dominate the input graph; but if the cross-attention is too sparse, the prompt graph will be limited to manipulating the input graph. We deem that a balance could be achieved by approximately equalizing the number of cross-connections with that of connections in the input graph, i.e., $n_e$. Therefore, we set the number of cross-connections to at most $n_e$ by connecting each node in the input graph with at most $\lfloor \frac{n_e}{a} \rfloor$ prompt tokens. Then, we can safely use a small $\delta_{cross}$ and cosine similarity $\frac{\mathbf{X}(i,:)\cdot\mathbf{P}_\theta^g(j,:)^\top}{\|\mathbf{X}(i,:)\|_2\|\mathbf{P}_\theta^g(j,:)\|_2}$ instead of $\sigma(\mathbf{X}(i,:)\mathbf{P}_\theta^g(j,:)^\top)$ to calculate the cross-connections. We demonstrate that our improved graph prompt works better in the later experiments.

## 4 MULTI-MODAL PROMPT LEARNING FOR GNNS

To adapt the GNN embeddings to the LLM's semantic embedding space and leverage the additional supervision provided by the text associated with graph labels, we explore the potential of multi-modal

---

[2]In fact, when executing the official implementation of [76] on Cora, the training loss does not decrease. Similar problems have been observed by another work [108].

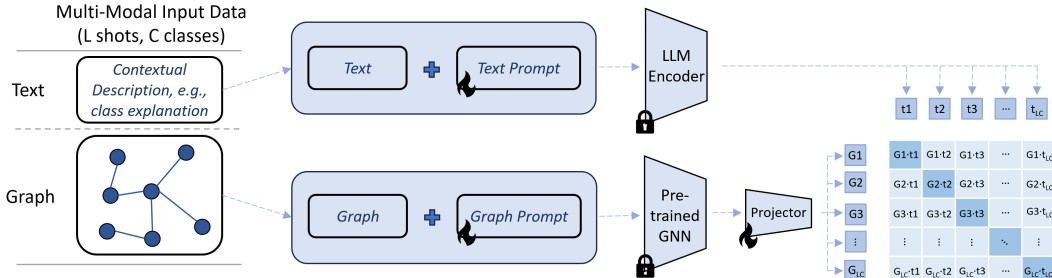

Figure 2: Similar to CLIP backbone, Morpher adapts the graph representations to semantic space through multi-modal prompt learning, even if the GNN and LLM are not jointly trained and are kept frozen.

prompt learning for both graphs and language. This approach is motivated by the intuition that only prompting on the graph data may limit the flexibility to adjust the LLM representation space. The overall paradigm of Morpher is illustrated in Figure 2. Given the data $\{(\mathcal{G}_i, t_i)\}_{i=1}^{L \times C}$, we aim to align graph embedding $\text{readout}(f_\phi^g(\mathcal{G}_i))$ with $\text{readout}(f_\phi^t(\text{Tokenize}(t_i)))$. Yet one direct issue is that, $\text{readout}(f_\phi^g(\mathcal{G}_i)) \in \mathbb{R}^{1 \times d_g}$ and $\text{readout}(f_\phi^t(\text{Tokenize}(t_i))) \in \mathbb{R}^{1 \times d_t}$ may have distinct dimensions. To address this issue, we adopt a cross-modal projector that learns to map the graph embedding space to the text embedding space. For an input $d_g$-dimensional vector $\mathbf{h}$, the projector maps it to a $d_t$-dimensional vector $\widetilde{\mathbf{h}}$:

$$\widetilde{\mathbf{h}} = \text{Proj}_\theta(\mathbf{h}) := \tanh(\mathbf{W}\mathbf{h} + \mathbf{b}) \in \mathbb{R}^{1 \times d_t} \tag{2}$$

As discussed in Sections 2 and 3, we introduce the text prompt $\mathbf{P}_\theta^t \in \mathbb{R}^{n_t \times d_t}$ with $n_t$ text prompt tokens and the graph prompt $\mathbf{P}_\theta^g \in \mathbb{R}^{n_g \times d}$ with $n_g$ graph prompt tokens. Let $\psi_g(\cdot, \mathbf{P}_\theta^g)$ be the graph prompting function, e.g., given any graph $\mathcal{G}$, the manipulated graph $\mathcal{G}_m = \psi_g(\mathcal{G}, \mathbf{P}_\theta^g)$.

Let $\omega_t(\cdot, \mathbf{P}_\theta^t)$ be the prompted text embedding given input text $t$. For the text prompt methods we choose, the prompted embedding is

$$\omega_t(t, \mathbf{P}_\theta^t) = [\mathbf{P}_\theta^t; f_\phi^t(\text{Tokenize}(t))]_{dim=0} \in \mathbb{R}^{(len(\text{Tokenize}(t)) + n_t) \times d_t} \tag{3}$$

Let $\omega_g(\cdot, \mathbf{P}_\theta^g)$ be the prompted graph embedding given input graph $\mathcal{G}$, then we have:

$$\omega_g(\mathcal{G}, \mathbf{P}_\theta^g) = f_\phi^g(\mathcal{G}_m) = f_\phi^g(\psi_g(\mathcal{G}, \mathbf{P}_\theta^g)) \in \mathbb{R}^{(n+n_g) \times d_g} \tag{4}$$

For the whole prompted text and the whole prompted graph, we apply readout (e.g., mean-pooling, max-pooling, etc.) to get their embedding:

$$e^t = \text{readout}(\omega_t(t, \mathbf{P}_\theta^t)) \in \mathbb{R}^{1 \times d_t}, \ e^{\mathcal{G}} = \text{readout}(\omega_g(\mathcal{G}, \mathbf{P}_\theta^g)) \in \mathbb{R}^{1 \times d_g} \tag{5}$$

For the given data $\{(\mathcal{G}_i, t_i)\}_{i=1}^{L}$, we compute the normalized embedding of prompted $\mathcal{G}_i$ and project it to the text embedding space through the projector:

$$z_i^{\mathcal{G}_{norm}} = \frac{e_i^{\mathcal{G}}}{||e_i^{\mathcal{G}}||_2} = \frac{\text{readout}(\omega_g(\mathcal{G}_i, \mathbf{P}_\theta^g))}{||\text{readout}(\omega_g(\mathcal{G}_i, \mathbf{P}_\theta^g))||_2}, \quad z_i^{\mathcal{G}} = \text{Proj}_\theta(z_i^{\mathcal{G}_{norm}}) \tag{6}$$

For the text embeddings, since for limited data the set $\mathcal{T} = \{t_i\}_{i=1}^{C}$ may contain texts that are semantically close as discussed in Appendix B.2, we extract a subspace in the text embedding space by normalizing the embedding as follows. We further normalize the text embeddings to the unit sphere, as standard practice in NLP.

$$\mu_t = \frac{1}{L} \sum_{i=1}^{L} \text{readout}(\omega_t(t_i, \mathbf{P}_\theta^t)), \quad e_{norm,i}^t = \text{readout}(\omega_t(t_i, \mathbf{P}_\theta^t)) - \mu_t \tag{7}$$

$$z_i^t = \frac{e_{norm,i}^t}{||e_{norm,i}^t||_2} = \frac{\text{readout}(\omega_t(t_i, \mathbf{P}_\theta^t)) - \mu_t}{||\text{readout}(\omega_t(t_i, \mathbf{P}_\theta^t)) - \mu_t||_2} \tag{8}$$

Finally, we use the in-batch similarity-based contrastive loss to train text prompts, graph prompts, and the projector as shown below, to adapt the pre-trained GNN representations to LLM.

$$\mathcal{L}_{G \to T} = -\frac{1}{B} \sum_{i=1}^{B} \log \frac{\exp(z_i^{\mathcal{G}} \cdot z_i^t / \tau)}{\sum_{j=1}^{B} \exp(z_i^{\mathcal{G}} \cdot z_j^t / \tau)} \tag{9}$$

Table 1: Few-shot graph classification performance (%). IMP (%): the average improvement (absolute value) compared to the **best result** among all the baseline methods.

| Training schemes | GNN pretraining | MUTAG | | ENZYMES | | PROTEINS | | MSRC_21C | |
|---|---|---|---|---|---|---|---|---|---|
| | | Acc | F1 | Acc | F1 | Acc | F1 | Acc | F1 |
| Supervised | N/A + GCN | 66.00 | 66.67 | 16.67 | 8.68 | 65.89 | 60.77 | 38.85 | 35.32 |
| | N/A + GAT | 66.00 | 65.69 | 16.45 | 4.65 | 64.75 | 64.08 | 41.14 | 39.86 |
| | N/A + GT | 66.66 | 66.26 | 15.62 | 4.22 | 62.81 | 57.12 | 38.28 | 41.62 |
| Pre-train + Fine-tune | GraphCL+GCN | 70.00 | 70.23 | 17.91 | 11.82 | 65.89 | 61.23 | 40.00 | 43.89 |
| | GraphCL+GAT | 70.00 | 69.73 | 17.91 | 10.46 | 65.16 | 63.92 | 44.57 | 45.74 |
| | GraphCL+GT | 68.00 | 67.81 | 17.70 | 8.99 | 63.28 | 56.41 | 41.71 | 43.73 |
| | SimGRACE+GCN | 66.67 | 67.27 | 17.29 | 8.78 | 66.82 | 64.70 | 40.57 | 43.84 |
| | SimGRACE+GAT | 70.67 | 69.10 | 16.87 | 7.18 | 65.42 | 63.65 | 42.85 | 42.37 |
| | SimGRACE+GT | 69.33 | 69.77 | 16.24 | 6.08 | 65.98 | 62.31 | 39.42 | 40.78 |
| AIO [76] | GraphCL+GCN | 64.67 | 39.27 | 17.50 | 4.97 | 61.35 | 44.93 | 3.59 | 10.09 |
| | GraphCL+GAT | 64.67 | 39.27 | 17.50 | 4.97 | 59.21 | 37.19 | 14.37 | 3.11 |
| | GraphCL+GT | 73.33 | 72.06 | 18.33 | 9.09 | 40.79 | 28.97 | 17.96 | 8.30 |
| | SimGRACE+GCN | 64.67 | 39.27 | 16.04 | 4.61 | 67.42 | 60.87 | 34.73 | 18.16 |
| | SimGRACE+GAT | 64.67 | 39.27 | 16.04 | 4.61 | 59.21 | 37.19 | 7.78 | 1.79 |
| | SimGRACE+GT | 36.00 | 27.26 | 17.50 | 8.15 | 50.56 | 49.34 | 32.34 | 15.13 |
| GPF-plus [19] | GraphCL+GCN | 68.67 | 67.27 | 16.88 | 15.48 | 64.75 | 61.45 | 47.42 | 29.02 |
| | GraphCL+GAT | 68.67 | 62.84 | 16.45 | 13.23 | 65.89 | 60.07 | 47.42 | 26.28 |
| | GraphCL+GT | 69.33 | 67.87 | 18.12 | 15.56 | 59.66 | 37.37 | 41.71 | 21.35 |
| | SimGRACE+GCN | 65.33 | 39.52 | 18.96 | 15.83 | 65.16 | 58.80 | 45.71 | 23.32 |
| | SimGRACE+GAT | 69.33 | 66.72 | 18.54 | 12.58 | 63.28 | 53.50 | 42.85 | 21.40 |
| | SimGRACE+GT | 70.00 | 67.31 | 17.91 | 14.69 | 64.83 | 52.97 | 34.13 | 20.13 |
| Gprompt [55] | GraphCL+GCN | 73.33 | 66.93 | 17.91 | 8.44 | 61.01 | 60.01 | 1.80 | 0.21 |
| | GraphCL+GAT | 64.67 | 62.63 | 17.08 | 14.18 | 50.56 | 50.55 | 1.80 | 0.22 |
| | GraphCL+GT | 70.67 | 70.02 | 17.91 | 9.64 | 63.28 | 58.65 | 1.80 | 0.21 |
| | SimGRACE+GCN | 65.33 | 39.52 | 17.29 | 14.48 | 52.70 | 52.68 | 1.80 | 0.21 |
| | SimGRACE+GAT | 67.33 | 65.88 | 16.25 | 11.31 | 59.10 | 58.72 | 1.80 | 0.21 |
| | SimGRACE+GT | 73.33 | 67.84 | 16.87 | 13.54 | 64.75 | 62.37 | 1.80 | 0.223 |
| Improved AIO (Ours) | GraphCL+GCN | 77.33 | 77.74 | 18.13 | 11.98 | 65.89 | 65.97 | 42.85 | 45.91 |
| | GraphCL+GAT | 74.67 | 75.51 | 18.33 | 11.26 | 65.76 | 66.05 | 46.85 | 51.39 |
| | GraphCL+GT | 74.67 | 74.67 | 19.16 | 9.04 | 68.12 | 68.18 | 42.85 | 43.54 |
| | SimGRACE+GCN | 68.00 | 69.01 | 17.91 | 9.02 | 66.82 | 66.40 | 44.57 | 49.24 |
| | SimGRACE+GAT | 77.33 | 77.20 | 18.75 | 9.39 | 66.91 | 65.49 | 45.14 | 42.31 |
| | SimGRACE+GT | 71.33 | 72.06 | 18.95 | 11.25 | 68.59 | 68.84 | 40.57 | 42.82 |
| Morpher (Ours) | GraphCL+GCN | 78.67 | 78.09 | 20.41 | 15.20 | 67.47 | 66.40 | 45.14 | 49.62 |
| | GraphCL+GAT | 79.33 | 79.15 | 23.12 | 18.01 | 70.89 | 70.30 | 50.85 | 54.48 |
| | GraphCL+GT | 76.00 | 76.51 | 19.58 | 13.28 | 73.53 | 72.48 | 45.71 | 48.41 |
| | SimGRACE+GCN | 69.33 | 70.27 | 19.79 | 14.94 | 67.10 | 66.15 | 45.71 | 51.24 |
| | SimGRACE+GAT | 78.00 | 77.65 | 20.21 | 16.27 | 68.12 | 67.26 | 45.71 | 51.13 |
| | SimGRACE+GT | 74.00 | 74.84 | 19.16 | 14.29 | 71.76 | 71.75 | 44.00 | 48.16 |
| IMP of ImprovedAIO | | 2.00 ↑ | 5.01 ↑ | 0.52 ↑ | 4.41 ↓ | 2.01 ↑ | 4.37 ↑ | 0.28 ↓ | 2.50 ↑ |
| IMP of Morpher | | 4.00 ↑ | 6.73 ↑ | 2.36 ↑ | 0.60 ↑ | 4.81 ↑ | 6.61 ↑ | 2.66 ↑ | 7.14 ↑ |

# 5 EXPERIMENTS

We evaluate our Morpher and the improved graph prompt through extensive experiments. In particular, we show that, compared to state-of-the-art baseline methods, they both more effectively adapt pre-trained GNNs to the specific downstream classification task, and introducing the text modality brings Morpher additional advantages over others. We use RoBERTa [53] as the LLM encoder for Morpher in the main experiments. We also validate the performance of Morpher with ELECTRA [12] and DistilBERT [68] in section 5.6 and Appendix C.3.

*Datasets.* We use real-world graph datasets from PyTorch Geometric [21], including one molecular dataset MUTAG [60]; two bioinformatic datasets ENZYMES and PROTEINS [5]; one computer vision dataset MSRC_21C [63]; three citation network datasets Cora, CiteSeer and PubMed [97]. We use real-world class names as text labels. More details are summarized in Appendix B.

*Pre-trained algorithms and GNN backbones.* To pretrain GNNs for evaluation, we adopt GraphCL [99] and SimGRACE [91] to pre-train three widely used GNN backbones: GCN [43], GAT [100] and GraphTransformer (GT) [42]. Additionally, in Appendix C.4, we verify the effectiveness of our methods on GNNs pre-trained using GraphMAE [27] and MVGRL [24], two other representative GNN self-supervised learning algorithms. For each dataset, to pre-train GNNs, we leverage self-supervised learning methods on all the graphs without any label information.

*Baselines and metrics.* We compare our methods with the following baselines: (1) training a GNN from scratch supervised by few-shot data (*"supervised"*); (2) fine-tuning a task head together with pre-trained GNN (*"fine-tune"*). We allow GNNs to be tunable for "*supervised*" and "*fine-tune*"; (3) state-of-the-art graph prompting algorithms: All-in-one (*"AIO"*) [76], which is the only graph prompting algorithm that supports multiple tasks in node-level, edge-level and graph-level to the best of our knowledge; GPF-plus [19] which prompt on graph features and Gprompt [55] which is based on subgraph similarity. We use accuracy and weighted F1 as classification performance metrics.

## 5.1 FEW-SHOT LEARNING

We investigate the ability of our improved graph prompt ("*ImprovedAIO*") and Multimodal prompt ("*Morpher*") to adapt frozen pre-trained GNNs using few-shot data. We focus on graph-level classification here and will further investigate the few-shot learning ability at other task levels in Section 5.2. Our few-shot learning setting is more challenging than existing works [76, 75] as we only allow no more than 10 labeled training and validation samples for each class. The results are shown in Table 1, where we report the average performance of 5 runs and calculate the absolute average improvement of our methods. From the results, given the same pre-trained GNN, our ImprovedAIO outperforms all the existing baseline methods except for ENZYMES F1 and MSRC_21C accuracy. Yet the performance of our ImprovedAIO on ENZYMES F1 and MSRC_21C accuracy is clearly better than those of the original AIO. Our Morpher can achieve an absolute accuracy improvement of 0.60% to 7.14% over the baselines across all datasets. Supervised by very limited labeled data, training a GNN from scratch is sub-optimal. Passing a GNN pre-trained on the dataset and fine-tuning it with a task head achieves sub-optimal but better results as the pre-trained GNN learns generalizable representations over the dataset through self-supervised learning. To mitigate the gap between the pre-training task and downstream tasks, AIO [76] proposes to learn graph prompts for downstream data. However, as we discussed in Section 3, when the node features are sparse vectors, the optimization would fail. Using the official implementation of AIO, we observe that the loss value tends to fluctuate, and the performance of AIO is usually even worse than supervised training. By restricting the cross-connections, our ImprovedAIO becomes more stable and constantly outperforms the fine-tuning baseline. Compared to the aforementioned methods, Morpher demonstrated superior performance due to its capability to adapt both graph and language representation spaces dynamically.

## 5.2 MORPHER SUPPORTS MULTIPLE-LEVEL TASKS

Inherited from AIO, our ImprovedAIO and Morpher also support adaptation to downstream tasks at node-level and edge-level, because they can be reformulated into graph-level tasks as discussed in Section 3. We demonstrate the performance of node classification and link prediction on Cora and CiteSeer. For node classification, we reformulate it to graph classification by inducing an ego-graph with 10 to 30 nodes centered at the node to classify. Each ego-graph has the same label as the center node. For edge classification, we randomly sample 200 edges from the graph, then create 200 negative samples by replacing one node in each edge. We label each graph according to whether it is a positive or negative sample.

Table 2: Node-level, edge-level performance.

| Dataset | | Cora | | CiteSeer | |
|---|---|---|---|---|---|
| Tasks | Methods | Acc | F1 | Acc | F1 |
| Node Level | Supervised | 52.83 | 47.73 | 63.91 | 64.82 |
| | Fine-tune | 56.37 | 55.04 | 64.87 | 66.42 |
| | AIO [76] | 14.69 | 7.10 | 18.93 | 6.92 |
| | ImprovedAIO | 58.46 | 55.10 | 66.44 | 66.53 |
| | Morpher | 61.26 | 62.36 | 68.20 | 68.56 |
| Edge Level | Supervised | 51.78 | 50.62 | 52.14 | 50.81 |
| | Fine-tune | 52.50 | 51.00 | 52.50 | 51.12 |
| | AIO [76] | 50.00 | 33.33 | 50.00 | 33.33 |
| | ImprovedAIO | 54.64 | 54.57 | 53.92 | 53.55 |
| | Morpher | 55.71 | 55.05 | 55.35 | 55.05 |

We use GraphCL+GCN to pre-train the GNN and report the mean performance in Table 2. The results are consistent with graph-level performance, where ImprovedAIO and Morpher outperform existing methods, with Morpher achieving slightly better performance than ImprovedAIO. Additionally, the training of the original AIO fails on both datasets due to the sparse node feature vectors.

## 5.3 DOMAIN TRANSFER

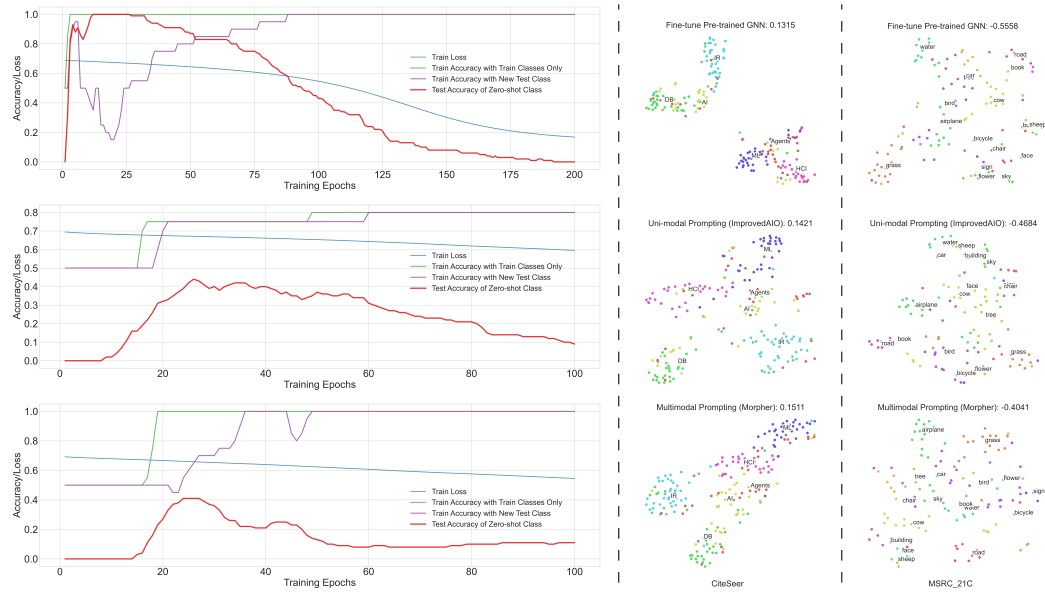

Figure 3: Results of novel class generalization (left); t-SNE embedding plots on CiteSeer, MSRC_21C (right). Train accuracy with train classes only is the accuracy of predicting the training graphs from the two training classes. Train accuracy with new test classes is the accuracy of predicting the training graphs from all three classes. Test Accuracy of zero-shot class is the accuracy of predicting the testing graphs from all three classes.

A key problem of the graph foundation model is whether we can adapt the pre-trained models to other data domains. Here, we explore the potential of using Morpher for such adaptation. We pre-train GNNs on ENZYMES or CiteSeer datasets, then test the classification performance on MUTAG and PubMed and report the results in Table 3. We unify the pre-train feature dimension with the downstream feature dimension by padding zeros or SVD reduction. From the results, Morpher demonstrates the best transferability, followed by ImprovedAIO. Also,

Table 3: Domain Transfer Performance.

| Target Domain | | MUTAG | | PubMed | |
|---|---|---|---|---|---|
| Target Task | | graph-level | | node-level | |
| Source | Methods | Acc | F1 | Acc | F1 |
| ENZYMES (graph-level) | Fine-tune | 68.00 | 55.04 | 47.57 | 36.07 |
| | ImprovedAIO | 70.67 | 64.07 | 50.28 | 50.51 |
| | Morpher | 72.67 | 73.29 | 54.42 | 53.96 |
| CiteSeer (node-level) | Fine-tune | 71.33 | 62.19 | 48.71 | 40.66 |
| | ImprovedAIO | 74.00 | 73.76 | 52.57 | 51.29 |
| | Morpher | 76.67 | 77.04 | 58.29 | 57.54 |

compared to the results on MUTAG in Table 1, all three methods have worse performances, because the GNNs are pre-trained on other datasets instead of MUTAG.

### 5.4 ZERO-SHOT CLASSIFICATION PROTOTYPE

An advantage of adapting pre-trained GNNs to the semantic embedding space is that GNNs might be empowered to "reasoning". Here, we conduct a novel experiment that generalizes GNN to an unseen class. Since no real-world data is available for this setting, we synthetically create three datasets, ZERO-Cora, ZERO-CiteSeer, and ZERO-PubMed, all from real-world connections. We aim to simulate a citation network with two research areas and an interdisciplinary research area in between. For each citation network, we randomly sample 120 nodes and induce their 2-hop ego-graphs, then replace the node features in 10 ego-graphs with $[1, 0]$ and another 10 ego-graphs with $[0, 1]$ to construct 20 training graph samples. For the remaining ego-graphs, we uniformly randomly replace the node features with $[1, 0]$ and $[0, 1]$ to construct 100 testing graph samples. We assign text labels of the first research area (e.g., "biology") to the $[1, 0]$ training graphs, the second research area (e.g., "informatics") to the $[0, 1]$ training graphs, and the interdisciplinary area (e.g., "bioinformatics") to the testing graphs. Intuitively, the nodes with feature $[1, 0]$ are papers in the first area, and other nodes with feature $[0, 1]$ are in the second area, which makes the datasets rational.

For each dataset, using GraphCL+GCN, we pre-train GNNs on all graphs. Then, we train Morpher on the training graphs, only knowing the text labels of the two training classes. Since we do not have validation data in zero-shot learning, we report the results of each epoch in Figure 3 (left). We observe that, while Morpher quickly adapts the GNN to downstream training data, the CLIP-like framework can predict the graphs in the novel class with good accuracy (red curve). Moreover, the training samples can be classified correctly from training and novel classes. Before the training

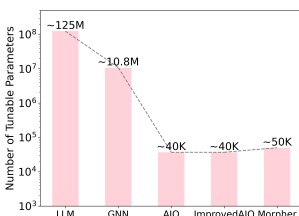 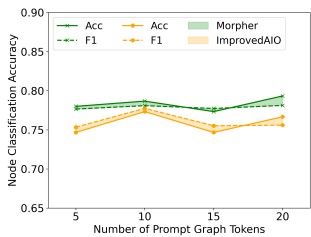 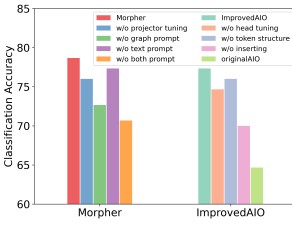

Figure 4: Efficiency comparison (left), parameter study (middle) and ablation study (right).

overfits, there is a period when Morpher can distinguish all the graphs from the training and novel classes with high accuracy.

Such zero-shot novel-class generalization ability validates Morpher's alignment between graph embeddings and text embeddings. When Morpher is trained on two classes of graphs with text labels of biology and informatics, a graph-in-the-middle will be classified as text-in-the-middle: bioinformatics, even if "bioinformatics" is an unseen label. The correspondence of in-the-middle graphs and texts shows the benefit and novelty of Morpher. To the best of our knowledge, this is the **first zero-shot classification prototype that generalizes GNN to unseen classes**.

### 5.5 EFFICIENCY AND EMBEDDING ANALYSIS

Without fine-tuning the GNN or LLM, the prompt-based methods have better parameter efficiency. As shown in Figure 4 (left), our ImprovedAIO and Morpher require similar numbers of parameters with AIO [76], which is $0.032\%$ to $0.46\%$ compared to either tune the LLM (RoBERTa) or GNN (GCN). Due to such parameter efficiency, our methods learn better graph representations given few-shot data. We visualize the graph embeddings of CiteSeer and MSRC_21C in Figure 3 and calculate the silhouette score, a metric for cluster quality ($\uparrow$) ranged in $[-1, 1]$. It turns out that our multimodal prompting leads to better adaptation.

### 5.6 HYPERPARAMETER AND ABLATION STUDY

We conduct the hyperparameter study by choosing and testing various numbers of graph prompt tokens for both ImprovedAIO and Morpher. The results are shown in Figure 4 (middle), from which we can observe that both methods are generally stable, and Morpher constantly outperforms ImprovedAIO under different choices. To verify the necessity of each component in our design, we compare Morpher and ImprovedAIO with multiple variants, respectively, and report the result in Figure 4 (right). We observe that removing any component would result in a performance drop. Additionally, our comparison of Morpher with ImprovedAIO throughout the experiments demonstrates that our multimodal design would lead to improvement over the uni-modal prompting of GNNs.

In the main experiments, we use RoBERTa as Morpher's text encoder. We also conduct experiments to verify the effectiveness of our proposed Morpher with ELECTRA [12] and DistilBERT [68] as the text encoder. Due to space limitation, we only show the F1 score of using ELECTRA in Figure 5, and more detailed experiment data can be found in Appendix C.3. In general, using ELECTRA and DistilBERT results in similar performance com-

Table 4: Effectiveness (F1 score) of Morpher with ELECTRA [12] as the text encoder.

| GNN pretraining | MUTAG | ENZYMES | PROTEINS | MSRC_21C |
|---|---|---|---|---|
| GraphCL + GCN | 78.17 | 15.79 | 65.66 | 47.19 |
| GraphCL + GAT | 75.75 | 11.37 | 65.66 | 49.01 |
| GraphCL + GT | 77.04 | 14.68 | 72.70 | 44.09 |
| SimGRACE + GCN | 70.99 | 12.41 | 67.77 | 48.44 |
| SimGRACE + GAT | 77.51 | 13.31 | 67.78 | 49.43 |
| SimGRACE + GT | 73.55 | 15.76 | 70.28 | 44.50 |

pared to using RoBERTa, showing the robustness of Morpher with respect to the language encoder.

As for the robustness with respect to the pre-trained GNNs, in the main experiments, we adopt two pre-train methods, GraphCL and SimGRACE to pre-train three different GNN architectures: GCN, GAT and GT. We further conduct experiments using GNNs pre-trained from GraphMAE [27] and MVGRL [24]. Due to the space limitation, we report the results and discuss in Appendix C.4.

### 5.7 MORPHER ON MOLECURENET

In this section, we demonstrate that, though not specifically designed for any downstream applications, the Morpher framework has the potential to be used in various downstream tasks, such as AI4Science

tasks. As for a case study, We use bace (inhibitors of human beta-secretase), tox21 (toxicology in the 21st century) and hiv (inhibit HIV replication) from MolecureNet [88]. These three datasets have 1513, 7831, and 41127 graphs to classify, respectively. In these datasets, each graph label is associated with a text description. The tasks on bace and hiv are bio-activity prediction and the task on tox21 is toxicity prediction. To adopt Morpher, we use GraphCL to pre-train the GAT model and initialize the text prompts and text labels using those from GIMLET [107].

Table 5: AUC-ROC ($\uparrow$) on MolecureNet (bace, tox21, hiv). Morpher-K denotes K shots.

| Dataset | KVPLM | MoMu | Galactica-1.3B | GIMLET-64M-50-shots | GAT-1M-supervised | Morpher-10 | Morpher-20 | Morpher-50 |
|---------|-------|------|----------------|---------------------|-------------------|------------|------------|------------|
| bace  | 0.5126 | 0.6656 | 0.5648 | 0.729 | 0.697 | 0.6231 | 0.6513 | 0.6858 |
| tox21 | 0.4917 | 0.5757 | 0.4946 | 0.652 | 0.754 | 0.6769 | 0.7275 | 0.7459 |
| hiv   | 0.6120 | 0.5026 | 0.3385 | 0.721 | 0.729 | 0.5742 | 0.7034 | 0.7283 |

KVPLM [101], MoMu [72], Galactica-1.3B [79] are zero-shot predictors for the three tasks; GIMLET-64M-50-shots is the GIMLET [107] model fine-tuned on 50 additional training samples[3]; GAT-1M-fully-supervised uses all the training data to train a GAT. Our Morpher-k-shots uses only k training samples. From the results, first, using only 10 training samples, Morpher can outperform the zero-shot baselines KVPLM, MoMu, and Galactica-1.3B. Second, using only 50 shots, Morpher can achieve similar performance with the fully supervised GAT. Third, using the same amount of few-shot data (50 shots), Morpher-50 outperforms GIMLET-64M-50-shots on tox21 and hiv, the two largest datasets among the three. This means our graph-text multi-modal prompt learning, with much fewer learnable parameters ($\sim 50K$), is more sample-efficient than fine-tuning language model encoder.

## 6 RELATED WORK

**GNN Pre-training.** Recently, a surge of graph pre-training strategies have emerged to address the issue of label scarcity in graph representation learning [29, 57, 75, 46, 39, 113]. The main idea of pre-trained graph models is to capture general graph information across different tasks and transfer this knowledge to the target task using techniques such as contrastive predictive coding [42, 20, 64, 91], context prediction [62, 30], prompt tuning [75, 19], and mutual information maximization [62, 73, 35]. For instance, [29] proposes to learn transferable structural information from three levels of graph topology, including node-level, subgraph-level, and graph-level. Different from these approaches, this paper aims to build up foundational GNNs by leveraging multi-modal prompt learning techniques.

**Graph Prompt Learning.** Prompting is now mainstream for adapting NLP tasks, and recent studies exploring prompt learning for GNNs mark a thriving research area [77, 87]. It is a promising way to adapt GNNs to downstream tasks through token-level [19, 78, 9, 75, 116] or graph-level [76, 32, 22] prompting. Among all the existing methods, All-in-one (AIO) [76] is the only algorithm to learn tunable graph prompts for node-level, edge-level or graph-level downstream tasks given few-shot labeled data (Table 8). Based on our improved AIO, we present a pioneer study to explore learning prompts in multiple modalities simultaneously while keeping the pre-trained models frozen.

**LLM on Graphs.** Inspired by the advances of large language models in NLP [111], researchers have begun to explore their potential for graph-related tasks [36]. Current approaches can be divided into two main categories. The first category employs LLMs as pre-trained feature extractors to enhance GNNs [16, 11, 115]. For example, GLEM [109] proposes to input the language representation as initial features for the GNN and train them iteratively. The second category focuses on integrating graph structures directly into LLM architectures [96, 104, 38]. A notable example is Patton [37], which pre-trains a joint architecture on text-attributed graphs. Despite these advancements, none of them have explored the collaboration between LLMs and GNNs under graph prompt learning.

## 7 CONCLUSION

In this work, we introduce Morpher, the first multimodal prompt learning paradigm that can semantically adapt pre-trained GNNs to downstream tasks with the help of LLM, while keeping both the pre-trained models frozen. To build Morpher, we first analyze the limitations of the state-of-the-art graph prompting technique and propose an improved version. Through extensive experiments, we demonstrate that our improved AIO can achieve outperformance, and our Morpher has further improvements in few-shot, multi-level task, or domain transfer settings. Additionally, using Morpher, we build the first GNN zero-shot classifier prototype that can be generalized to novel testing classes.

---

[3]the performance of GIMLET and other baselines are directly from the GIMLET paper [107].

# 8 ETHICS STATEMENT

There are no ethical concerns associated with this research. The datasets and related resources used in this study are publicly accessible and have been widely employed in the existing works.

# 9 REPRODUCIBILITY STATEMENT

To ensure reproducibility of this work, We provide the experiment code in the supplementary materials, which can be executed on a medium-powerful machine. We provide well-written README and configuration files in order to reproduce our results. We also discuss the experiment environment in detail in Appendix C.1. We use benchmark datasets that are available to the public. The experiment environments, including the details of the machine we used, are discussed in Appendix C.2. We explicitly stated the amount of memory and time needed for execution.

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

# A PROOF OF THEOREM 3.1

*Proof.* The cross-entropy loss between the true distribution $p(\cdot)$ and the predicted distribution $q(\cdot)$ is given by:

$$\text{CE}(p, q) = -\sum_y p(y) \log q(y)$$

where $q(y) = Pr(c(\mathbf{x}) = y)$.

To find the optimal classification, we minimize the cross-entropy loss subject to the constraint $\sum_y q(y) = 1$. We define the Lagrangian as:

$$\mathcal{L}(q, \lambda) = -\sum_y p(y) \log q(y) + \lambda \left( \sum_y q(y) - 1 \right)$$

For any $y \in \mathcal{Y}$, take the derivative of $\mathcal{L}$ with respect to $q(y)$ and $\lambda$ and set them to zero, we get:

$$\frac{\partial \mathcal{L}}{\partial q(y)} = -\frac{p(y)}{q(y)} + \lambda = 0$$

$$\frac{\partial \mathcal{L}}{\partial \lambda} = \sum_y q(y) - 1 = 0$$

Solving these equations, we find:

$$q(y) = \frac{p(y)}{\lambda}$$

$$\sum_y q(y) = \sum_y \frac{p(y)}{\lambda} = \frac{1}{\lambda} \sum_y p(y) = 1$$

Therefore, $\lambda = 1$ and $q(y) = p(y)$.

Thus, the optimal classification is $Pr(c(\mathbf{x}) = y) = p(y)$.

$\square$

# B DATASET DETAILS

## B.1 DATASET STATISTICS

Table 6 summarizes the statistics of the public real-world datasets, which we used in the few-shot experiments. For our synthetic datasets in the zero-shot prototype, we summarize their statistics in Table 7. As discussed in Section 5.4, the connections of our synthetic datasets are real, and we only replace the node feature by $[1, 0]$ and $[0, 1]$. The code to download the public data and the code to create synthetic data are provided in the supplementary materials.

## B.2 TEXT LABELS

When created, real-world graph datasets are usually coupled with textual meanings, but a common practice is to convert the textual meanings into numbers to create labels, which weakens the super­vision of the graph data. For each real-world dataset, we convert the numerical labels back to text labels and feed into Morpher Language encoder through "[learnable text prompt] + [text label]". The mapping from the numbers to text labels for each dataset are provided as follows:

Table 6: Dataset statistics

| Dataset | task level | # graphs | average # nodes | average # edges | # feature dimension | # classes | # shots per class | feature characteristic |
|---|---|---|---|---|---|---|---|---|
| MUTAG | graph | 188 | 17.9 | 39.6 | 7 | 2 | 10 | one-hot, sparse |
| ENZYMES | graph | 600 | 32.6 | 124.3 | 3 | 6 | 10 | one-hot, sparse |
| PROTEINS | graph | 1113 | 39.1 | 145.6 | 3 | 2 | 10 | one-hot, sparse |
| MSRC_21C | graph | 209 | 40.28 | 96.60 | 22 | 17 | 1 | one-hot, sparse |
| Cora | node, edge | 1 | 2708 | 10556 | 1433 | 7 | 2 (node), 20 (edge) | sum 1, sparse |
| CiteSeer | node, edge | 1 | 3327 | 9104 | 3703 | 6 | 2 (node), 20 (edge) | sum 1, sparse |
| PubMed | node | 1 | 19,717 | 88648 | 500 | 3 | 10 | TF-IDF value, dense |

Table 7: Synthetic Zero-shot Class Generalization Dataset statistics

| Dataset | # graphs | average # nodes | average # edges | #feature dimension | # classes | # shots per class |
|---|---|---|---|---|---|---|
| ZERO-Cora | 120 | 8.41 | 10.38 | 2 | 2 | 10 |
| ZERO-CiteSeer | 120 | 10.03 | 21.31 | 2 | 2 | 10 |
| ZERO-PubMed | 120 | 20.33 | 41.75 | 2 | 2 | 10 |

**MUTAG.** MUTAG is a dataset of nitroaromatic compounds, aiming to predict their mutagenicity on Salmonella typhimurium. Therefore, the mapping from numerical labels to text labels is: {0: non-mutagenic on Salmonella typhimurium, 1: mutagenic on Salmonella typhimurium}.

**ENZYMES.** ENZYMES aims to predict which subcategory each enzyme belongs to. The subcategories are: 0: oxidoreductases, 1: transferases, 2: hydrolases, 3: lyases, 4: isomerases, 5: ligases.

**PROTEINS.** PROTEINS is a dataset comprising proteins classified as either enzymes or non-enzymes. Therefore, the mapping is: 0: 'enzyme', 1: 'non-enzyme'.

**MSRC_21C.** Each graph in MSRC is constructed according to an image. The graph label is the image label. MSRC_21C contains 20 classes in MSRC, and "C" here means "Challenging" as the graphs(images) that are easy to classify has been filtered. The mapping from the numerical labels to text labels is: {0: building, 1: grass, 2: tree, 3: cow, 4: sheep, 5: sky, 6: airplane, 7: water, 8: face, 9: car, 10: bicycle, 11: flower, 12: sign, 13: bird, 14: book, 15: chair, 16: road}.

**Cora.** Cora is a citation network of papers in seven research areas. Each paper is labeled according to its corresponding research area. The mapping from the numerical labels to text labels is: {0: case based, 1: genetic algorithms, 2: neural networks, 3: probabilistic methods, 4: reinforcement learning, 5: rule learning, 6: theory}.

**CiteSeer.** CiteSeer is a citation network of papers, each labeled according to one of six research areas. The mapping from the numerical labels to text labels is: {0: Agents, 1: AI, 2: DB, 3: IR, 4: ML, 5: HCI}. We note that using abbreviations of the research area is not an issue because these abbreviations frequently appear, and the LLM tends to tokenize each of them as one token.

**PubMed.** PubMed is a collection of scientific publications from the PubMed database related to diabetes, classified into one of three categories. The mapping from the numerical labels to text labels is: {0: Diabetes Mellitus Experimental, 1: Diabetes Mellitus Type 1, 2: Diabetes Mellitus Type 2}.

**Edge-level tasks.** Cora, CiteSeer and PubMed can also be used as link prediction datasets. For link prediction, the mapping from the numerical labels to text labels is: {0: not connected, 1: connected}.

**Synthetic Zero-shot Class Generalization Datasets.** For ZERO-Cora, we synthetic three classes of ego-graph in a citation network. The first and second classes, respectively, have text labels "machine learning" and "theory", and the third (novel) class to generalize is "machine learning

Table 8: Comparison of graph prompts.

| Method | prompt level | level of supported downstream tasks | | | learnable prompt | semantic |
|---|---|---|---|---|---|---|
| | | node-level | edge-level | graph-level | | |
| GPF-Plus [19] | token-level | √ | × | × | √ | × |
| Gprompt [55] | token-level | √ | × | √ | √ | × |
| VNT [78] | token-level | × | × | √ | √ | × |
| ULTRA-DP [9] | token-level | √ | × | × | √ | × |
| GPPT [75] | token-level | √ | × | × | √ | × |
| SGL-PT [116] | token-level | √ | × | × | √ | × |
| SAP [22] | graph-level | √ | × | √ | √ | × |
| PRODIGY [32] | graph-level | √ | √ | √ | × | × |
| All-in-one (AIO) [76] | graph-level | √ | √ | √ | √ | × |
| ImprovedAIO (ours) | graph-level | √ | √ | √ | √ | × |
| Morpher (ours) | graph-level | √ | √ | √ | √ | √ |

theory". For ZERO-CiteSeer, we synthetic three classes of ego-graph in a citation network. The first and second classes, respectively, have text labels "biology" and "informatics", and the third (novel) class to generalize is "bioinformatics". For ZERO-PubMed, we synthetic three classes of ego-graph in a citation network in the medical domain. The first and second classes, respectively, have text labels "cardiology" and "neurology", and the third (novel) class to generalize is "neurocardiology".

## C  Experiment Details

### C.1  Reproducibility

**Code.**  The code for the experiments is provided in the supplementary material with a well-written README file. We also provide the commands and instructions to run the code. The datasets used will be automatically downloaded when the code is executed.

**Environment.**  We run all our experiments on a Windows 11 machine with a 13th Gen Intel(R) Core(TM) i9-13900H CPU, 64GB RAM, and an NVIDIA RTX A4500 GPU. We have also tested the code on a Linux machine with NVIDIA TITAN RTX GPU. All the code of our algorithms is written in Python. The Python version in our environment is 3.9.18. In order to run our code, one has to install some other common libraries, including PyTorch, PyTorch Geometric, pandas, numpy, scipy, etc. Please refer to our README in the code directory for downloading instructions.

We have optimized our code and tested that the space cost of **the CPU memory is less than 16 GB, and the space cost of the graphics card is less than 6 GB**. The execution time to run an experiment is less than 20 minutes on our machine.

### C.2  Implementation Details

We provide the configuration files for the experiments to reproduce the results. We initialize the graph prompt using kaiming_initialization, and we initialize the text prompts through real token embeddings. We have tested multiple initializations, and they would not affect the overall results. Specifically, we initialize the text prompt for each dataset as follows.

MUTAG: "a graph with property"; ENZYMES: "this enzyme is"; PROTEINS: "this protein is"; MSRC_21C: "an image of"; Cora: "a paper of"; CiteSeer: "a paper of"; PubMed: "a paper of"; Edge tasks: "central nodes are".

In our few-shot setting, we split the labeled data into training samples and validation samples at approximately 1:1. For all the parameters, we used the Adam optimizer, whose learning rate and weight decay are provided in the configuration files.

## C.3 EXPERIMENT WITH ELECTRA AND DISTILBERT

On the LLM pre-training side, RoBERTa is one of the most advanced encoder-only LLMs until now, and we have demonstrated the effectiveness with RoBERTa serving on the LLM side in the Morpher paradigm. Additionally, we conducted experiments with ELECTRA [12] and DistilBERT [68]. Using these two LLMs, Morpher can also achieve comparable performances to RoBERTa. The results are shown as follows.

Table 9: Few-shot graph classification performance (%) of Morpher with ELECTRA [12] as language encoder. Other experiment settings are identical to the main experiment.

| GNN pretraining | MUTAG | | ENZYMES | | PROTEINS | | MSRC_21C | |
|---|---|---|---|---|---|---|---|---|
| | Acc | F1 | Acc | F1 | Acc | F1 | Acc | F1 |
| GraphCL + GCN | 78.00 | 78.17 | 20.41 | 15.79 | 67.38 | 65.66 | 43.42 | 47.19 |
| GraphCL + GAT | 76.67 | 75.75 | 20.41 | 11.37 | 66.26 | 65.66 | 44.57 | 49.01 |
| GraphCL + GT | 76.67 | 77.04 | 19.16 | 14.68 | 73.06 | 72.70 | 42.28 | 44.09 |
| SimGRACE + GCN | 70.00 | 70.99 | 19.79 | 12.41 | 68.96 | 67.77 | 45.71 | 48.44 |
| SimGRACE + GAT | 77.33 | 77.51 | 18.12 | 13.31 | 68.96 | 67.78 | 44.00 | 49.43 |
| SimGRACE + GT | 72.67 | 73.55 | 18.33 | 15.76 | 70.18 | 70.28 | 41.14 | 44.50 |

Table 10: Few-shot graph classification performance (%) of Morpher with DistilBERT [68] as language encoder. Other experiment settings are identical to the main experiment.

| GNN pretraining | MUTAG | | ENZYMES | | PROTEINS | | MSRC_21C | |
|---|---|---|---|---|---|---|---|---|
| | Acc | F1 | Acc | F1 | Acc | F1 | Acc | F1 |
| GraphCL + GCN | 78.00 | 78.61 | 20.62 | 10.00 | 66.44 | 65.54 | 43.42 | 47.98 |
| GraphCL + GAT | 77.33 | 75.64 | 21.25 | 15.87 | 70.59 | 68.25 | 45.14 | 48.82 |
| GraphCL + GT | 74.67 | 75.20 | 19.58 | 14.96 | 70.27 | 70.55 | 44.57 | 47.28 |
| SimGRACE + GCN | 69.33 | 70.36 | 20.62 | 18.82 | 66.91 | 66.41 | 45.14 | 47.77 |
| SimGRACE + GAT | 77.33 | 76.90 | 18.54 | 14.44 | 67.56 | 65.08 | 45.71 | 44.36 |
| SimGRACE + GT | 72.67 | 73.52 | 17.91 | 11.06 | 70.55 | 70.36 | 45.14 | 44.01 |

In general, using ELECTRA and DistilBERT results in similar performance compared to using RoBERTa, showing the robustness of Morpher with respect to the language encoder.

## C.4 EXPERIMENT WITH GNNs TRAINED USING GRAPHMAE AND MVGRL

In the main pages, we used GraphCL and SimGRACE to show that Morpher achieves better performance given a pre-trained GNN. Additionally, to further verify the robustness of Morpher over the pre-train method, we conducted experiments on the pre-trained GNNs using GraphMAE [27] and MVGRL [24]. We use GCN as the GNN backbone and RoBERTa as the LLM encoder, and the results are reported as follows.

Table 11: Few-shot graph classification performance (%) of Morpher with the GNN pre-trained by GraphMAE [27]. Other experiment settings are identical to the main experiment.

| GNN pretraining | MUTAG | | ENZYMES | | PROTEINS | | MSRC_21C | |
|---|---|---|---|---|---|---|---|---|
| | Acc | F1 | Acc | F1 | Acc | F1 | Acc | F1 |
| Pre-train + Fine-tune | 71.33 | 71.41 | 16.04 | 12.14 | 65.86 | 65.22 | 39.42 | 40.20 |
| ImprovedAIO | 76.67 | 76.95 | 19.58 | 12.59 | 66.36 | 65.30 | 42.28 | 46.81 |
| Morpher | 78.67 | 78.67 | 20.20 | 16.95 | 67.38 | 65.66 | 45.71 | 48.49 |

Table 12: Few-shot graph classification performance (%) of Morpher with the GNN pre-trained by MVGRL [24]. Other experiment settings are identical to the main experiment.

| GNN pretraining | MUTAG | | ENZYMES | | PROTEINS | | MSRC_21C | |
| --- | --- | --- | --- | --- | --- | --- | --- | --- |
| | Acc | F1 | Acc | F1 | Acc | F1 | Acc | F1 |
| Pre-train + Fine-tune | 68.67 | 69.46 | 16.45 | 10.16 | 65.15 | 64.71 | 38.85 | 40.56 |
| ImprovedAIO | 74.67 | 74.00 | 18.13 | 15.57 | 66.54 | 65.90 | 42.85 | 46.66 |
| Morpher | 78.00 | 77.81 | 18.96 | 14.97 | 67.56 | 66.79 | 44.57 | 48.67 |

Using GraphMAE or MVGRL to pre-train the GNN, the trend of performance is similar to that when using GraphCL or SimGRACE. Also, ImprovedAIO and Morpher's performance is similar to that of pre-trained GNNs from GraphCL or SimGRACE and can still significantly outperform the pre-train + fine-tune baseline, showing the robustness of Morpher with respect to the pre-training strategy.

## D  LIMITATIONS

Graph prompt learning assumes the "pre-train + prompt" framework to build graph foundation models, yet there could be other paths to achieve graph-related foundation models. Also, graph prompt learning only works on the graph neural network architecture, and might not work for other architectures that are proposed in the future. Another limitation of this work is the requirement of language encoder. While RoBERTa is one of the most advanced encoder-only language models and can be considered an LLM with over 0.1B parameters, more recent LLMs such as Llama or Mistral cannot be used in Morpher because they are decoder-only LLMs and do not explicitly have an encoder. Yet it is possible to retrieve the hidden representation before the decoder layer. We leave this direction as future work.

