# OpenReview forum: "Multi-modal Prompt Learning Empowers Graph Neural Networks with Semantic Knowledge"
_ICLR.cc/2025/Conference — ICLR 2025 Conference Withdrawn Submission_

### Official Review · Reviewer_Kgke · 2024-11-01

**Soundness:** 3
**Presentation:** 3
**Contribution:** 3
**Rating:** 5
**Confidence:** 4

**Summary:**

This paper introduces a multimodal prompt learning method called Morpher that uses both graph and text prompts to align graph representations with LLM’s semantic embeddings. This method is able to do zero-shot classification for unseen graph classes. In their paradigm, the pre-trained GNN and the LLM are kept frozen, making it much more efficient than finetuning. They also introduce an improved graph prompting method that cross-connections between the prompt graph and the input graph share the same density with the input graph. By using a projection layer and training with contrastive loss, they align the graph and text prompt embeddings in the same space, allowing for effective adaptation to various tasks with limited downstream data.

**Strengths:**

1. Morpher is able to adapt to various downstream tasks with limited labels. It is also able to do zero-shot classification on unseen classes.
2. They introduce an improved method of graph prompting without overwhelming or limiting the input graph's features.
3. The use of a contrastive loss function to align the graph and text embeddings helps bridge the gap between graph structure and semantic text features, enhancing the model's flexibility across modalities.

**Weaknesses:**

1. The effectiveness of Morpher relies on the availability and quality of semantic labels in the text associated with graph data. When these labels are noisy, incomplete, or inconsistent, the alignment might be weakened and the performance of downstream tasks might be reduced.
2. Freezing GNN and LLM reduces the number of trainable parameters, but it also limits adaptation flexibility. Certain tasks might require fine-tuning the GNN or LLM to capture task-specific information.
3. The independently pre-trained GNN and LLM may cause huge representation gaps between the two modalities, causing difficulty in the alignment process.
4. Applying the graph prompting might be computationally expensive on large and complex graphs.

**Questions:**

1. How does different text labels affect the model performance? Does randomizing the class text label lead to comparable performance in few-shot setting? How fine-grained text can the model take? Will adding explanations to the text label help improve the performance?
2. Why does the test accuracy of zero-shot class decrease as the training epoch increases?
3. What’s the distribution of graph and text representation? Does there still exist domain shift after the alignment? Can you visualize the distributions of graph and text representations before and after alignment, using t-SNE or UMAP?

---

> ### Author Response · Authors · 2024-11-30
> **Thanks for reviewing**
>
> We sincerely thank the reviewer. We would like to take this opportunity to briefly address your concerns for clarification.
>
> 1. The consideration of noisy labels is beyond the scope of this research. In this study, all labels used have been manually verified to ensure their correctness.
> 2. We chose to freeze the GNN and LLM components primarily for efficiency. Our experiments demonstrated that tuning only the prompts from both modalities provides sufficient flexibility to adapt to downstream out-of-domain tasks, including AI4Science applications. For those with greater computational resources, the GNN and LLM can be set to train mode instead of evaluation mode, allowing for further fine-tuning if desired.
> 3. This paper specifically addresses the challenges inherent in the alignment process between modalities, which is a central focus of our contributions.
> 4. While the original AIO framework requires establishing all connections between the prompt graph and input graph, we acknowledge that this approach can become computationally expensive for large graphs. To address this, our improved AIO and Morpher frameworks employ sparse connections, which enhance training stability while reducing computational overhead.
> 5. We utilized real text labels associated with each graph dataset. For example, in the Cora dataset, the numerical labels are mapped to text labels as follows:
>    {0: case based, 1: genetic algorithms, 2: neural networks, 3: probabilistic methods, 4: reinforcement learning, 5: rule learning, 6: theory}. These text labels represent the research areas of the publications in the citation network.
> 6. Because the classifier overfits the seen labels as the training goes on.
> 7. We have visualized the t-SNE embedding plot in Figure 3.

---

### Official Review · Reviewer_M9xE · 2024-11-01

**Soundness:** 3
**Presentation:** 2
**Contribution:** 1
**Rating:** 3
**Confidence:** 5

**Summary:**

This paper proposes a new framework for integrating pre-trained GNNs with LLMs, leveraging the reasoning capabilities of LLMs alongside the structural representation advantages of GNNs. Specifically, the authors prompt both the LLM and the GNN with trainable parameters while keeping the parameters of both base modules frozen. This prompting, along with a cross-model projector, assists in aligning the embeddings produced by the GNN and the LLM for an input instance, such as a graph. Additionally, they introduce a variant of a current GNN prompting method that addresses its shortcomings through an attention mechanism.

**Strengths:**

- The proposed method does not involve training the base GNN or LLM, which reduces the number of parameters needed for training and enhances complexity and versatility.
- The authors reasonably justify the shortcomings of one current state-of-the-art model and propose modifications to mitigate its issues, empirically demonstrating the improvements achieved by these modifications.
- The main part of the prompting method, Morpher, is explained clearly with formulations, making it easy to understand.
- The authors provide sufficient and clear visualizations for their proposed methods.

**Weaknesses:**

- The authors do not compare their work with state-of-the-art approaches that combine LLMs and GNNs, such as [1]
- While the authors claim to provide a theoretical analysis that "in many cases, state-of-the-art graph prompt is unable to learn good representations," this analysis is limited to Lemma 3.1. This lemma presents a trivial conclusion that does not significantly contribute to the development of the method or elaborate on the effectiveness of current prompting methods. It may be beneficial to remove this lemma and its proof, as it has been widely discussed in the literature on GNNs, and cite related works [2], which would help streamline the paper.
- In the Background section, few-shot prompt learning is only addressed in the context of methods that learn trainable parameters. It would be valuable to also discuss methods that do not train parameters, particularly regarding the intrinsic in-context learning of these methods.
- Some definitions and formulations appear extraneous. For instance, Equation 2 addresses concepts that are reiterated in Equation 6 and relate to matching dimensionality. The term $\tilde{h}$ does not appear in subsequent equations, and similarly, the introduction of $e^t$ in Equation 5 is later replaced by $e^t_{norm,i}$ without further context. Streamlining these sections could reduce unnecessary complexity.
- The authors do not discuss the inference process of their method at test time. It would be helpful to clarify whether the same model must be used at inference time and whether the LLM or the GNN makes the final prediction.
- The claim that this method is the first to address zero-shot classification for GNNs to unseen classes may overlook existing approaches. Referencing works such as [1] and [3] and comparing with them would provide necessary context.
- While the authors mention generating textual features from numerical features of the original datasets, it appears that for node classification datasets, only the labels of the nodes are converted to text, as indicated in Appendix B.2, rather than the node features themselves. Moreover, converting high-dimensional feature vectors to text features is not trivial and could exceed the acceptable input length for LLMs. This oversight may limit the model's generalizability to node/edge-level tasks due to the neglect of node features.


[1] Hao Liu, Jiarui Feng, Lecheng Kong, Ningyue Liang, Dacheng Tao, Yixin Chen, & Muhan Zhang (2024). One For All: Towards Training One Graph Model For All Classification Tasks. In The Twelfth International Conference on Learning Representations.

[2] Keyulu Xu, Weihua Hu, Jure Leskovec, & Stefanie Jegelka (2019). How Powerful are Graph Neural Networks?. In International Conference on Learning Representations.

[3] Chen, J., Mi, R., Wang, H., Wu, H., Mo, J., Guo, J., Lai, Z., Zhang, L., & Leung, V. (2024). A Review of Few-Shot and Zero-Shot Learning for Node Classification in Social Networks. IEEE Transactions on Computational Social Systems, 1-15.

**Questions:**

- In Section 3, it is stated that the number of cross-connections is controlled to $n_e$ by thresholding. How are the best edges selected? Is this selection random, or is there a specific criterion?
- The objective function used for training Morpher does not incorporate labels or the task but focuses solely on aligning the prompting parameters for the GNN and LLM. This raises concerns about potential misalignment with pre-trained task-specific information, as merely aligning representations without consideration of the task could lead to forgetting previously learned knowledge. What is the rationale behind this objective function, and how does it mitigate this issue?
- Given that the objective function does not involve labels and the GNN and LLM have frozen parameters, how is the model evaluated in a few-shot setting with labels? Does this imply that the base GNN and LLM are pre-trained, with prompting applied afterward without using labels? If so, the contributions to domain transfer and zero-shot learning may be limited, as the fundamental learning from one domain to another occurs prior to prompting. The proposed method may primarily align representations of the LLM and GNN, which has already been explored in few-shot settings within a single domain.

---

> ### Author Response · Authors · 2024-11-30
> **Thanks for reviewing**
>
> We sincerely thank the reviewer. We would like to take this opportunity to briefly address your concerns for clarification.
>
> 1. We did not compare our method with One for All because it operates under significantly more advantaged settings. Specifically, One for All assumes input graphs with rich text-attributed features, whereas our work primarily utilizes limited textual information derived from node labels. This fundamental difference in assumptions makes a direct comparison impractical and potentially misleading.
> 2. Lemma 3.1 was included to substantiate the claim that AIO struggles with learning good representations due to training instability. While we acknowledge the point that this conclusion is relatively straightforward, we believe its inclusion is necessary for completeness.
> 3. Due to strict page limitations, we prioritized content directly related to the main text. However, we appreciate the suggestion to include non-trainable few-shot prompt learning methods and intrinsic in-context learning.
> 4. Equation 2 is necessary to define what Proj() is to be later used in Equation 6. $\tilde{h}$ does not have to appear in subsequent equations because it’s just a note for a general vector to show how Proj() modifies dimensionality: h = Proj(h) = tanh(Wh + b). e^t is also a general notation to help readers understand $e^t_{norm, i}$ for the given data {(Gi, ti)}.
> 5. Our inference process is similar to the CLIP framework, which relies on similarity-based predictions. This allows both the GNN and LLM components to work in tandem without additional fine-tuning during inference.
> 6. Please refer to 1.
> 7. We neither converted nor planned to convert the node features into text, as we only have limited text labels available rather than detailed paragraphs of text typically associated with text-attributed graphs.
> 8. The best edges are selected based on the highest attention scores.
> 9. Labels are implicitly incorporated into our objective function through the z vectors.
> 10. Same as 9. This is a misunderstanding, and we did incorporate labels in the objective function.

---

### Official Review · Reviewer_9HU4 · 2024-11-03

**Soundness:** 2
**Presentation:** 2
**Contribution:** 2
**Rating:** 5
**Confidence:** 4

**Summary:**

This paper introduces a new paradigm for enhancing Graph Neural Networks (GNNs) through multi-modal prompt learning, which leverages text data to align downstream tasks with pre-trained GNNs using only a few semantically labeled samples. The approach embeds graphs in the same semantic space as large language models (LLMs) by jointly learning graphs and text prompts. Building on theoretical insights, the authors improve existing graph prompt methods and propose a multi-modal framework that fully utilizes knowledge from pre-trained models without requiring fine-tuning of either the GNN or LLM. This setup significantly reduces the number of learnable parameters. Extensive experiments on real-world datasets demonstrate the paradigm's effectiveness across few-shot, multi-task, and cross-domain settings. Additionally, the framework includes a zero-shot classification prototype that enables GNNs to generalize to previously unseen classes.

**Strengths:**

S1. The manuscript is well-organized and clearly written, with structured and detailed mathematical derivations. For instance, in the "Improved Graph Prompt Design" section, the paper rigorously derives the formulation for balancing cross-connections between input nodes and prompt nodes.
S2. The proposed sparse cross-connection mechanism in Morpher prevents prompt features from overwhelming graph features, offering a balanced integration for improved feature alignment.
S3. Comprehensive experiments across diverse datasets, including Cora, CiteSeer, and synthetic zero-shot datasets, effectively demonstrate Morpher's adaptability and robustness.
S4. Results show that Morpher achieves superior performance over baselines in few-shot and zero-shot tasks, validating its cross-domain generalization capabilities.

**Weaknesses:**

W1. This paper proposes a sparse cross-connection mechanism for Morpher to prevent prompt features from overloading graph features, but could this also limit information exchange between graph and text modalities, weakening alignment quality? For complex tasks requiring richer feature fusion, this limited interaction may be insufficient.

W2. Morpher’s reliance on graph-text fusion assumes complementary information from both modalities, but in tasks with weaker associations, might this integration introduce noise and reduce performance?

W3. The paper does not analyze how differences in data distribution between pre-training and target datasets influence transfer performance, which is crucial for domain adaptation. Quantifying these differences (e.g., via JS or KL divergence) could clarify Morpher’s sensitivity to dataset similarity.

W4. The domain transfer experiment only tests on limited datasets (MUTAG and PubMed), which share similar characteristics in molecular biology. It is unclear if the findings generalize to more diverse domains like social networks.

W5. In the "Zero-Shot Classification Prototype" section, the zero-shot experiment relies on synthetic datasets (ZERO-Cora, ZERO-CiteSeer), which are generated by modifying features and labels and may fail to capture the complexity of natural data distributions. I wonder about Morpher’s generalization performance on real-world datasets.

W6. The author mentions "Figure 5" in the text, though no such figure exists; it seems they meant to refer to "Table 4" instead.

W7. Some relevant works are missing, e.g., Killing Two Birds with One Stone: Cross-modal Reinforced Prompting for Graph and Language Tasks, KDD 2024.
Natural Language Is All a Graph Needs. arxiv. 2023

**Questions:**

Please refer to the above weaknesses.

---

> ### Author Response · Authors · 2024-11-30
> **Thanks for reviewing**
>
> We sincerely thank the reviewer. We would like to take this opportunity to briefly address your concerns for clarification.
>
> 1. This is out of the scope of this research. Even with the original full connection, there is no direct information exchange between the graph and text representations.
> 2. The consideration of noisy labels is beyond the scope of this research. In this study, all text labels have been manually verified to ensure correctness.
> 3. The primary goal of our domain transfer experiment is to demonstrate that our paradigm outperforms direct fine-tuning in cross-domain scenarios.
> 4. Please refer to 3.
> 5. To the best of our knowledge, no real-world datasets currently meet the semantic label requirements for our zero-shot experiment.
> 6. Yes. It should be Table 4.
> 7. We will check those references.

---

### Official Review · Reviewer_3FNq · 2024-11-03

**Soundness:** 2
**Presentation:** 2
**Contribution:** 2
**Rating:** 5
**Confidence:** 3

**Summary:**

This paper proposes Morpher, which leverages the text modality to align downstream tasks and data with any pre-trained GNN given only a few semantically labeled samples. The key idea is to embed the graphs directly in the same space as the LLM by multi-modal prompt learning. Experiments demonstrate the superior performance of Morpher in few-shot, multi-task-level, and cross-domain settings.

**Strengths:**

1.  The proposed Morpher can generalize GNN to unseen classes.
2.  The authors provide codes for reproducibility.
3.  This paper is easy to follow.

**Weaknesses:**

1. The authors only show that AIO [76] is unable to learn good representations of the downstream data. In my opinion, some state-of-the-art graph prompt methods such as PRODIGY [Ref1] have addressed this issue.
2. From Appendix D.3 in [Ref1], PRODIGY can also apply to the zero-shot classification. Therefore, the comparison of PRODIGY and Morpher is necessary.
3. Some references such as [Ref2, Ref3] are missing.



[Ref1] PRODIGY: Enabling In-context Learning Over Graphs. NeurIPS 2023.

[Ref2] Harnessing Explanations: LLM-to-LM Interpreter for Enhanced Text-Attributed Graph Representation Learning. ICLR 2024.

[Ref3] Label Deconvolution for Node Representation Learning on Large-scale Attributed Graphs against Learning Bias. TPAMI 2024.

**Questions:**

See Weaknesses.

---

> ### Author Response · Authors · 2024-11-30
> **Thanks for reviewing**
>
> We sincerely thank the reviewer. We would like to take this opportunity to briefly address your concerns for clarification.
>
> PRODIGY does not perform strict zero-shot learning. The zero-shot evaluation described in Section D.3 ("For our evaluation process, we computed zero-shot transfer performance of the model on the test set") refers to directly applying the pre-trained model to downstream tasks. However, in these downstream tasks, k-shot data samples for each label are incorporated into the prompt/context to predict the query sample (PRODIGY paper page 7, Experiments, Evaluations, “we construct a k-shot prompt for test nodes (or edges) from the test split by randomly selecting k examples per way from these available examples”).
>
> We will check the suggested references.

---

### Author Response · Authors · 2024-11-30

We sincerely thank the reviewers. Unfortunately, we have decided to withdraw this submission. Nonetheless, we have addressed the concerns raised in each review to provide clarification and ensure public transparency.

---

### Note · Authors · 2024-11-30

**Comment:**

We sincerely thank the reviewers. Unfortunately, we have decided to withdraw this submission. Nonetheless, we have addressed the concerns raised in each review to provide clarification and ensure public transparency.

**Withdrawal Confirmation:**

I have read and agree with the venue's withdrawal policy on behalf of myself and my co-authors.